# Genome-Wide Association Study Identifies Candidate Genes Related to the Linoleic Acid Content in Soybean Seeds

**DOI:** 10.3390/ijms23010454

**Published:** 2021-12-31

**Authors:** Qin Di, Angela Piersanti, Qi Zhang, Cristina Miceli, Hui Li, Xiaoyi Liu

**Affiliations:** 1Research Center of Integrative Medicine, School of Basic Medical Sciences, Guangzhou University of Chinese Medicine, Guangzhou 510006, China; di.qin@unicam.it; 2Innovative Institute for Plant Health, Zhongkai University of Agriculture and Engineering, Guangzhou 510225, China; zhangqi18922@163.com; 3School of Biosciences and Veterinary Medicine, University of Camerino, 62032 Camerino, Italy; angela.piersanti@unicam.it (A.P.); cristina.miceli@unicam.it (C.M.)

**Keywords:** soybean, LA, genome-wide association study, RNA-seq, MALDI-TOF IMS, *WRI1*

## Abstract

Soybean (*Glycine max* (L.) Merrill) oil is a complex mixture of five fatty acids (palmitic, stearic, oleic, linoleic, and linolenic). The high content of linoleic acid (LA) contributes to the oil having poor oxidative stability. Therefore, soybean seed with a lower LA content is desirable. To investigate the genetic architecture of LA, we performed a genome-wide association study (GWAS) using 510 soybean cultivars collected from China. The phenotypic identification results showed that the content of LA varied from 36.22% to 72.18%. The GWAS analysis showed that there were 37 genes related to oleic acid content, with a contribution rate of 7%. The candidate gene *Glyma.04G116500.1* (*GmWRI14*) on chromosome 4 was detected in three consecutive years. The *GmWRI14* showed a negative correlation with the LA content and the correlation coefficient was −0.912. To test whether *GmWRI14* can lead to a lower LA content in soybean, we introduced *GmWRI14* into the soybean genome. Matrix-assisted laser desorption/ionization time-of-flight imaging mass spectrometry (MALDI-TOF IMS) showed that the overexpression of *GmWRI14* leads to a lower LA content in soybean seeds. Meanwhile, RNA-seq verified that *GmWRI14*-overexpressed soybean lines showed a lower accumulation of *GmFAD2-1A* and *GmFAD2-1B* than control lines. Our results indicate that the down-regulation of the *FAD2* gene triggered by the transcription factor *GmWRI14* is the underlying mechanism reducing the LA level of seed. Our results provide novel insights into the genetic architecture of LA and pinpoint potential candidate genes for further in-depth studies.

## 1. Introduction

Soybeans (*Glycine max* (L.) Merrill) are a major oilseed crop that provides edible oil and protein in China [1]. Soybean oil is a complex mixture of five fatty acids: palmitic acid (C16:0), stearic acid (C18:0), oleic acid (C18:1), linoleic acid (LA) (C18:2), and linolenic acid (C18:3) [2]. The fatty acid composition of soybean oil ranges from about 15% to 33% oleic acid, 43% to 56% linoleic acid, 5% to 11% linolenic acid, and 11% to 26% saturated acids acid [3]. As a food commodity, soybean oil has a less-than-desirable fatty acid composition [4]. The quality of soybean oil is determined by its fatty acid composition. Linolenic acid can lower the cholesterol content in the blood [5,6], but it is not resistant to high temperatures, is not easy to preserve, and lowers the nutritional value of soybean oil [7,8,9]. LA belongs to the class of polyunsaturated fatty acids and is known to have beneficial properties for health [9]; however, the oxidative stability values of oil enriched with LA show a poor stability during a frying cycle [10]. Soybean oil with a high LA content has a higher oxidation rate [11]. Decreasing the amount of polyunsaturated fatty acids present through lowering the LA content improves a plant oil’s oxidative stability and brings its acyl composition closer to that of olive oil. As a result, the cultivation of soybean varieties with a low LA content has become an important goal of high-quality soybean breeding [12].

The biosynthesis of LA is highly regulated, involving the spatial separation of biosynthetic steps between different organelle compartments and the careful control of several biosynthetic steps through one or multiple biochemical mechanisms. Based on next-generation sequencing technology and genome-wide association analysis (GWAS), considerable research has gone into genetically reducing the level of LA, and some key genes involved in the development of LA have been identified [13,14,15]. For instance, fatty acid desaturases 2 (*FAD2*) is a key gene encoding an ER membrane-bound FA desaturase 2, which catalyzes the conversion of oleic acid to LA [16,17]. The RNA interference (RNAi) of *FAD2* and b-ketoacyl-acyl carrier protein synthase (*KASII*) decrease the accumulation of LA [18]. Fatty acid desaturase-3 (*FAD3*) genes that are responsible for the conversion of LA precursors to linolenic fatty acid precursors have been characterized [19]. Research showed that the loss of function mutations seen in *FAD3* resulted in the accumulation of 67% LA in seed oil [20,21].

*WRINKLED1* (*WRI1*) encodes an APETALA2/ethylene responsive element binding protein (AP2/EREBP). WRI1 is the key transcription factor found to directly regulate fatty acid synthesis [22,23]. Further studies have shown that *WRI1* plays a role in processes such as lipid assembly, storage, seed development, and photosynthesis [24,25,26]. For example, the overexpression of the maize *ZmWRI1* expressed under the embryo-preferred *OLE* promoter increased the fatty acid content of maize kernels [27]. In addition, the introduction of a seed-specific expression cassette carrying the *Arabidopsis* transcription factor *AtWRI1* into soybean led to seed oil with levels of palmitic acid of up to approximately 20% [28]. Clearly, WRI1 acts as a key regulator of oil biosynthesis, which, when expression is perturbed, translates to changes in the accumulation of different kinds of fatty acids [29,30].

Genome-wide association analysis (GWAS) represents a powerful tool for discovering (single-nucleotide polymorphisms) SNPs associated with complex traits [31]. GWAS has become an affordable and powerful tool for dissecting complex traits in soybean [32,33,34]. Prior to this, GWAS has been performed for the dissection of soybean fatty acids [35]. As factored spectrally transformed linear mixed models (fastlmmc model) run an order of magnitude faster than other efficient algorithms, the fastlmmc model was used for GWAS in many studies [1,2]. In our study, we discovered a candidate gene *GmWRI14* that was correlated with LA content by GWAS analyses in three years. *GmWRI14* belongs to the plant WRI1 protein family. The overexpression plasmid DNA was transferred into the soybean cultivar Jinong38 (JN38) by agrobacterium-mediated transformation. As a result, we produced novel soybean lines containing only 11% LA content in the seed. The present study could serve as a good reference for future studies on high-quality soybean breeding.

Specifically, 510 soybean cultivars from China (Heilongjiang Province, Shandong Province, and Guangdong Province) were collected as natural populations. The soybean lines were planted in a field belonging to Guangzhou University from 2018 to 2020. Specific-Locus Amplified Fragment sequencing (SLAF-seq) technology was used to sequence the genomes of 510 soybean materials, while GWAS was conducted to find the genomic regions associated with LA. In addition, a total of 37 candidate genes were identified related to LA. A new candidate gene related to LA was discovered by GWAS in three years consistently.

## 2. Results

### 2.1. Phenotypic Variation of LA Content in Soybean Seeds

From 2018 to 2020, 510 soybean lines with different fatty acid contents were collected from three different regions and determined by NIRSTM DS 2500 (FOSS, Hillerod, Denmark) (Figure 1A). The variation range of LA was 35.12–71.28%, the standard deviation (SD) of the LA content of soybean seeds was 8.2, and the LA content of seeds approached the normal distribution (Figure 1B). The results indicated that the LA content of different soybean lines was significantly different, which was in accordance with the genetic law of quantitative traits. We used the linear regression model to analyze the relationship between LA and oleic acid. According to the linear regression model, the content of LA was significantly negatively correlated with the content of oleic acid (Figure 1C). The correlation coefficient between oleic acid and LA was −0.750.

### 2.2. A New Candidate Gene Related to LA Was Discovered by GWAS in 2018–2020

In our experiments, SLAF-seq technology was used to sequence soybean genomic DNA. A total of 2,423,512 SNP markers were obtained for genetic mapping; the results of the SNP distribution on chromosomes are shown in Figure 1D. The threshold value was set at −log_(*p*) _ >  4.20 (red). A total of 612 SNPs of LA were detected on chromosomes 1, 4, 9, 13, 14, 15, 16, 19, and 20. Among these, more than 50% of the SNPs were located in the intergenic regions (the stretch of DNA sequences located between genes). In total, 4.98% of the SNP loci were located in protein coding regions. Additionally, 6.34% of the SNP markers were located in introns (Figure 1E). We found that most of the SNPs on chromosome 4 were exactly located at the coding sequence coordinates (CDS): 116,500–122,890. Population structure analysis is a cluster analysis method that is currently widely applied; it is helpful for understanding the evolutionary process of materials based on SNPs. This experiment was used to analyze the soybean population structure based on 2,423,512 SNPs. We analyzed the data with EIGENSTRAT in order to study 510 soybean lines. It was concluded that the samples we collected can be represented as an admixture of two ancestral populations (Appendix A).

Based on the LA content of 510 soybean lines, the fastlmmc model was used for GWAS. SNP markers that were significantly correlated with the LA were detected. The LD distance was set to 8.9 kb and candidate genes related to LA were screened within the LD distance. Manhattan maps showing the LA content across different years (2018–2020) are shown in Figure 1F. In 2018, 17 candidate genes related to soybean LA content were screened using GWAS (Appendix A), 10 candidate genes were screened by GWAS in 2019 (Appendix A), and 10 candidate genes were screened in 2020 (Appendix A). The functional prediction of candidate genes is also shown in Appendix A. The first promising candidate gene, *Glyma.04G116500.1*, was detected by GWAS in three years consistently (Figure 1F). The amino acid sequence encoded by *Glyma.04G116500.1* was compared with *AtWRI1* (Gene ID:824599), the homology of the amino acid sequences between Glyma.04G116500.1 and AtWRI1 is 62.34%. A multiple alignment showed that Glyma.04G116500.1 contained two (AP2/EREB) DNA-binding domains, which were located at positions 54 to 220 of the amino acid sequence (Appendix A). Therefore, it is speculated that *Glyma.04G116500.1* may belong to the AP2/EREBP family. There is no functional report about the candidate genes in the soybean database. According to Swissport annotation, *Glyma.04G116500.1* belongs to the plant WRI1 family. The *Glyma.04G116500.1* is located on chromosome 4, we named *Glyma.04G116500.1* as *GmWRI14*.

### 2.3. The Expression of Candidate Gene GmWRI14 in Different Tissues and Developmental Stages of Soybean

To gain further insights into the organ-specific expression profiles of *GmWRI14*, the expression of the *GmWRI14* gene was investigated by qRT-PCR. The results showed that the highest transcript abundance of *GmWRI14* was found in soybean seeds. In order to validate its association with LA content, the *GmWRI14* expression was measured in different tissues (root, stem, leaf, and seed). The *lectin* gene (GenBank: A5547-127) was used as a reference gene. The qRT-PCR results showed that the *GmWRI14* in soybean seedlings was expressed in different tissues, but the relative expression was significantly different, ranging from 26.12 to 52.32 in seeds, from 0.87 to 4.62 in leaves, from 11.12 to 28.25 in stems, and from 16.60 to 31.12 in roots (Appendix A). As a general conclusion, we can say that the correlation coefficient between *GmWRI14* expression level and LA content is −0.901~−0.912 (*p* < 0.01) (Appendix A). This result strongly indicates that the candidate gene *GmWRI14* is closely related to the LA content, specifically showing a negative effect on the LA content.

### 2.4. The Generation and Molecular Characterization of Transgenic Soybean Plants Over-Expressing GmWRI14

The recombinant plasmid designated pCAMBIA3300-*GmWRI14* was introduced into the *Agrobacterium tumefaciens* strain (Appendix A). About 480 soybean cotyledon calluses were subjected to transformation. A Southern blot assay was used to detect the presence and determine the copy number of the *GmWRI14* gene in the six selected putative transgenic lines, with one copy of *GmWRI14* being detected in the T0/T1 generation of each line (Figure 2A,B). The full-length original blots are included in the additional files (Appendix A). The fatty acid content of the T2 generation transgenic soybean seeds was determined using a near-infrared grain analyzer. The data revealed the trend of increase in the total oil content in T2 seeds from plants grown under greenhouse conditions. The LA distribution in seeds was detected using matrix-assisted laser desorption/ionization time-of-flight imaging mass spectrometry (MALDI-TOF IMS); however, a significant decrease in LA was detected by MALDI-TOF-MS. The LA content in the positive strain decreased from 52.10% to 11.42% and the oleic acid content increased from 23.21% to 41.02% compared to the content in the JN 38 recipient (Appendix A). Meanwhile, the LA content of the *GmWRI14* transgenic lines was reduced under field conditions during 2021. The result indicated that *GmWRI14* decreases the LA content in soybean seed. The transgenic lines and control varieties were investigated under field conditions. Both the control JN38 and transgenic lines had white flowers, round leaves, and gray hairs. There was no significant difference between the transgenic lines and control varieties. The *GmWRI14* gene was highly up-regulated in transgenic soybeans compared to JN38, and its expression has been found to be closely connected to oil accumulation in various soybean plants. It has been proven that the overexpression of *GmWRI14* can lead to a significant increase in the oil content (Appendix A). Meanwhile, the average value of the results of three 100 grain weight measurements showed that there was an obvious difference in the 100 grains weight due to the different densities owing to the different contents of LA (Appendix A). Our result demonstrates the usefulness of the *GmWRI14* gene for high-quality soybean breeding.

### 2.5. The Reduction in LA Triggered by GmWRI14 Expression Is Due to the Down-Regulation of Soybean FAD2

To understand the genetic underpinnings of the variation in LA in soybean seeds, we first analyzed the differentially expressed genes (DEGs) between *GmWRI14* transgenic soybeans and control JN38 using the RNA-Seq data. RNA from three biological replicates of six transgenic lines and control JN38 was sequenced; as a result, more than 5.1 × 10^7^ clean reads were obtained for different soybean lines after removing low-quality reads, with the error rate of clean reads ranging from 0.02 to 0.03. A total of 1542 DEGs were screened by transcriptomics after a Kyoto Encyclopedia of Genes and Genomes (KEGG) analysis. The functional information of DEGs was found to contain graphical representations of cellular processes, such as LA metabolism and alpha-linolenic acid metabolism (Figure 3C). In the LA metabolism, a differentially expressed gene Volcano Plot (DEGVP) analysis revealed eight differentially expressed genes to be associated with LA metabolism (Figure 3B). To identify the *FAD2* genes present in the soybean genome down-regulated by *GmWRI14*, we analyzed the expression of *FAD2* genes in transgenic soybeans using qRT-PCR to determine whether the expression of this gene was altered. The *lectin* gene (GenBank: A5547-127) was used as a reference gene. As a result, the expression of *GmFAD2-1A* and *GmFAD2-1B* genes in transgenic soybean lines was reduced and the *GmFAD2-1A* gene was found to be expressed in different tissues. However, the relative expression was significantly lower, ranging from 1.33 to 5.31 in the transgenic soybean leaves, from 10.21 to 19.23 in stems, and from 17.32 to 23.22 in seeds (Table 1). Meanwhile, the relative expression of the *GmFAD2-1B* gene was significantly lower, ranging from 3.14 to 5.12 in soybean stems, from 5.46 to 9.11 in leaves, and from 21.21 to 27.35 in seeds (Table 1). As a general conclusion, the correlation coefficient between *GmFAD2-1A* and *GmWRI14* is −0.970–−0.982 (*p* < 0.01) and the correlation coefficient between *GmFAD2-1B* and *GmWRI14* is −0.880–−0.814 (*p* < 0.01) (Table 1). This result strongly indicates that the candidate gene *GmWRI14* plays a negative role in regulating the LA content in the seeds. The LA content triggered by *GmWRI14* expression is due to the down-regulation of soybean *FAD2* (*GmFAD2-1A* and *GmFAD2-1B*).

## 3. Discussion

The LA is the shortest-chain n-6 fatty acid and the most common polyunsaturated fatty acid (PUFA) in plant oils and can be present in commercial oils at levels >50% [36]. Overall, 60–70% of fatty acids in soybean oil are unsaturated; they are mainly characterized by high LA contents, which are responsible for the low oxidative stability of the oil. High temperatures cause the oxidative polymerization of LA. Soybean oil has a less than desirable fatty acid composition [37]. Low LA soybeans have a competitive agronomic yield potential, with many candidate genes associated with LA and discovered by GWAS already having been reported [13,14].

*WRINKLED1* (*WRI1*), an APETALA2 (AP2)-type transcription factor, has been shown to be needed for the regulation of carbon partitioning into fatty acid synthesis in plant seeds [38]. In 2018, the introduction of the *Arabidopsis* transcription factor *AtWRI1* into soybean plant led to an increasing level of palmitate up to approximately 20% [28]. In another study, the close correlation between WRI1 and Stearoy-Acyl-Carrier-Protein Desaturase (*SAD*) expression suggested the regulatory role of WRI1 in LA accumulation [39]. In our study, a new soybean WRINKLED 1 transcription factor *GmWRI14* associated with LA in soybean seeds was detected by GWAS. The *GmWRI14* is closely related to the LA content, specifically showing a negative effect on the LA content in soybean seeds. In one study, with the overexpression of *AtWRI1*, the levels of LA decreased from 42.1 ± 0.3% to 4.6 ± 0.3% in transgenic *Arabidopsis* plants [40]. In our study, transgenic soybean lines harboring *GmWRI14* displayed a 5-fold decrease in LA compared to the control soybean line JN38. Under the control of the seed-specific *napin* promoter, *GmWRI1* led to oil accumulation in soybean seed [41]. Similar results were reported from our research, with the overexpression of *GmWRI14* in soybean increasing the oil content in soybean seed. To identify *GmFAD2-1A* and *GmFAD2-1B* genes present in the soybean genome down-regulated by *GmWRI14*, a qRT-PCR analysis was conducted. These results indicate that *GmWRI14* specifically down-regulates *GmFAD2-1A* and *GmFAD2-1B* in transgenic soybean plants, leading to a decrease in LA.

In previous research, candidate SNPs and genes significantly associated with LA were screened by GWAS. The functions of the candidate genes involved in LA metabolism and regulation, such as transcription activator-like effector nucleases (TALENs), acetyl-CoA carboxylase biotin carboxylase subunit, glycosyltransferase group 1, and *FAD2*, were identified [16,17,18]. In our work, GWAS was used to find potential SNPs and genes correlated with the LA content, with most SNPs on chromosome 4 being found to be exactly located at CDS region 11,65,00–12,28,90, the region identified for the LA is colocalizing with previously studied [1,2]. The candidate gene *GmWRI14* was detected in different years. We introduced *GmWRI14* into soybean, translated it to an oil with low LA, and noticed discernable changes in the total oil content under a greenhouse or field environment. The transgenic soybean lines also showed the requirement for both seed-specific homologues of *FAD2* in soybean to be down-regulated to achieve a lower LA content to be verified. The overexpression of *GmWRI14* led to the accumulation of more C18:1 and less C18:2, creating soybean oil with a high oleic acid/linoleic acid ratio. This is the first time that WRI1 had been reported to be associated with LA content in soybean. Our results provide a basis for deciphering the mechanisms underlying the determination of fatty acid composition. The widespread limitation of soybean oil production is due to the rapid development of rancidity during the storage of oil, leading to the generation of off flavors [42,43,44]; therefore, it is necessary to adopt a stabilization step to decrease the content of polyunsaturated fatty acids in soybean seed.

Since WRI1 has been shown to be required for the regulation of carbon partitioning into fatty acid synthesis in plant seeds, many studies to date have focused on identifying genes upregulated by WRI1; several target genes, including *Arabidopsis* hemoglobin 1 (*AtGLB1*) and Glycerol-3-Phosphate Acyltransferase (*GPAT*), have been found. In our study, many rate-limiting genes involved in glycolysis, PPP, and FA synthesis and desaturation, such as SAD, BS, MCMT, TAL, and *FAD2*, showed an expression correlation with *GmWRI14* (Figure 4). In *A. thaliana*, the WRINKLED transcription factor is known to activate genes that regulate fatty acid biosynthesis in seeds via binding the AW box (Figure 4). While these genes explain some facets of the *WRI1* overexpression phenotype, they do not fully account for the *WRI1* function, suggesting that additional target genes remain to be discovered. Meanwhile, the repressor functions of *WRI1* have been poorly investigated. In this study, we demonstrated a transcriptional repressor function of *GmWRI14*, showing that *GmWRI14* repressed the *FAD2*, which has not been reported in previous studies. Our results significantly improve our understanding of the transcriptional control exerted by *GmWRI14* (Figure 4). The lack of an AW-box-related cis element in the promoter of *FAD2* suggests that the *GmWRI14*-mediated regulation of this gene might be indirect. Thus, it is reasonable to speculate that another trans-acting factor (depicted as “X” in Figure 4) that is specifically associated with repressed target genes converts *GmWRI14* into a repressor. Although we mainly focused on the selected target gene in the current study, many potentially interesting target genes are repressed by *GmWRI14*. An interesting topic for future studies would be testing the relative contribution of the co-repressor functions of *GmWRI14* in *A. thaliana* models.

## 4. Materials and Methods

### 4.1. The Plant Materials

In total, 510 soybean materials were provided by the Biotechnology Center of Jilin Agricultural University and Guangzhou University of Chinese Medicine. The soybean materials were planted in 2017–2018 in Changchun, China (123.53° E, 23.84° N); in 2018–2019 in Qingdao, China (120.41° E, 36.39° N); and in 2019–2020 in Guangdong, China (113.41° E, 23.31° N). A randomized complete block design was used. The field was divided into three blocks and those were subdivided into eight sections. During late R8 soybean growth stages, the seeds were harvested and then the harvested seeds were used for phenotyping for the LA content.

### 4.2. Determination of Fatty Acids in Soybean Seeds

The content of LA and four other fatty acids (stearic acid, palmitic acid, oleic acid, and linolenic acid) in 510 soybean cultivars seeds was determined by NIRSTM DS 2500 (FOSS, Hillerod, Denmark). Fresh seeds of the R8 stages were ground into a fine powder in liquid nitrogen and we put the powder into the instrument. The near-infrared spectroscopy mathematical model of soybean fatty acid content was provided by the Beijing Biomarker Biotechnology Company, PR China. The LA distribution in the seed was detected using MALDI-TOF-MS. The matrices were prepared according to the layer method of the instrument maker’s manual using 9-aminoacridines (C_13_H_10_N_2_). The Matrix (1 μL) was spotted onto an MSP 96 target polished steel (Bruker Daltonics, Bremen, Germany) and allowed to air dry for approximately 5 min at room temperature. Each sample (1 μL) was diluted in 50 μL of matrix, and 1–1.5 μL of the sample/matrix mixture was deposited on the top of the matrix thin layer and then dried at room temperature. LA were measured on a MALDI Microflex LT instrument equipped with a 60 Hz nitrogen laser (Bruker Daltonics, Bremen, Germany). Mass spectra were recorded in the linear positive mode and externally calibrated using an LA standard. To increase the detection sensitivity, the following conditions were used: mass range 34,211–52,571 Da, sample rate 1.00 GS/s, laser shots 100, laser power 85%, laser frequency 60, and detector gain 33X. Each extracted sample was analyzed at least three times. The SPSS version 22.0 software (SPSS Inc., Chicago, IL, USA) was used to calculate the correlation coefficient of fatty acid.

### 4.3. Genotyping of Soybean Germplasms

The total genomic DNA was extracted from the leaves of each soybean line using the CTAB method according to Murray and Thompson [45]. The 510 soybean materials were genotyped by SLAF-seq and SNP molecular markers were developed [1]. The sequencing service was provided by Beijing Biomarker Biotechnology Company, PR China. SNP molecular markers were used for GWAS analysis and a population structure analysis. The restriction endonuclease combination was *RsaI-HaeIII.* Population structure analysis can be used to quantify the number of ancestors of the studied population and infer the source of each sample. Based on the SNPs, the EIGENSOFT version 5.0 software (Broad Institute of MIT and Harvard, Cambridge, MA, USA) was used to assess the population structure.

### 4.4. The Genome-Wide Association Analysis (GWAS)

Based on the 2,423,512 SNP markers obtained by SLAF-Seq technology, all the SNP markers obtained from genotyping were used for GWAS. The TASSEL software can calculate the Q matrix of sample population structure according to the K matrix and finally obtain a correlation value for each SNP maker. The threshold value was set at −log_(*p*)_  >  4.20. In this experiment, Manhattan maps were constructed using the Haploview software (BROAD Inc., Chicago, IL, USA). The Manhattan map was used to represent the correlation between genotypic data and phenotypic data, and correlation values between SNP markers and LA content were obtained. In this study, the fastlmmc model was used for GWAS and the candidate genes were predicted using the Swiss-Prot and NR databases. We evaluated the genome-wide LD in 510 accessions and found that the LD (R^2^) values decayed to half of the maximum value within 8.9 kb. Using 8.9 kb as the linkage disequilibrium attenuation distance, candidate genes related to soybean linoleic acid traits were screened within the LD distance.

### 4.5. Quantitative Reverse Transcription-PCR

The total RNA was extracted using the Eastep^®^ Super total RNA extraction Kit (TaKaRa, Vero Beach, FL, USA), then cDNA synthesis was performed using a reverse transcription kit (Omega, Norcross, GA, USA). The qRT-PCR analysis was performed using a Bio-Rad CFX system (Amersham Biosciences, Little Chalfont, Buckinghamshire, UK). Gene-specific primer pairs P3: (5′-TTGCCTGTCTAGATCCACAGCTGGTACCGAT-3′) and P4: (5′-TTGTGACCTCGACCTATTGGCGTTACCAATT-3′) were used to amplify *GmWRI14*. The *lectin* gene (GenBank: A5547–127) was used as the reference gene. The reference gene was amplified with primer pairs P5: (5′-GCACTTAAGATACTCTAGGTAC-3′) and P6: (5′-CCACCTCCCTACTATCCATT-3′). The amplification reaction conditions were pre-denaturation at 95 °C for 10 min, with denaturation at 95 °C for 10 s, annealing at 53 °C for 20 s, and extension at 72 °C for 15 s. The amplification reaction conditions for the gene *GmWRI14* were pre-denaturation at 95 °C for 10 min, denaturation at 95 °C for 30 s, annealing at 59 °C for 30 s, extension at 72 °C for 35 s, 35 cycles, and extension at 72 °C for 10 min. Three biological replicates were used for each gene.

### 4.6. Vector Construction and Plant Transformation

The 1815 bp cDNA of the *GmWRI14* gene from the cultivar JN38 was ligated into the *BamHI-SacI* site of pCAMBIA3300 to place the coding region under the regulatory control of the 35S promoter and nos terminator. The recombinant plasmid was named pCAMBIA3300- *GmWRI14*, and then the recombinant plasmid was introduced into the *Agrobacterium tumefaciens* strain. We introduced *GmWRI14* into the soybean cultivar JN38 (Approval number 2012010) using the *Agrobacterium tumefaciens* strain LBA4404 [46]. The transformation process was divided into five sequential steps: bacterial inoculation, cocultivation, resting, selection, and plant regeneration. The callus tissue used was originally derived from the cotyledon-nodes of soybean. The seed of JN38 possesses good agronomic characteristics with a normal LA content. T0 plants (primary transformants) established in the green house that grew normally, flowered, and set seeds.

### 4.7. The RNA-Seq Library Preparation and Sequencing

The plant samples were processed for total RNA extraction using the Eastep^®^ Super total RNA extraction kit (TaKaRa, Vero Beach, FL, USA). The RNA quality was checked using a Nanodrop 2000c (Thermo Scientific, Hudson, OH, USA). RNA-seq library preparation and sequencing were performed using the protocols described previously [47,48,49]. The fold change for gene expression was calculated by normalizing Ct values at each developmental stage against an endogenous control (Gmβ-actin: *Gm15g05570)* using the 2^−ΔΔCt^ method [50].

### 4.8. Data Analysis

The phenotypic data were measured and recorded using the Microsoft Excel 2020 software. Each data point was based on three replicates. A differential saliency analysis, analysis of variance, correlation analysis, and descriptiveness analysis were performed using the SPSS 22.0 (SPSS Inc., Chicago, IL, USA) software [50]. The statistical significance at *p* ≤ 0.01 was calculated. The histograms were constructed using the Graphpad Prism software (Graphpad Company, San Diego, CA, USA).

## 5. Conclusions

In this study, a genome-wide association study (GWAS) was conducted on a panel of 510 germplasm resources. We combined the GWAS and RNA-seq methods to identify candidate genes for LA metabolism in soybean seeds. A new candidate gene, *GmWRI14*, was discovered by GWAS in three consecutive years. The RNA-seq results indicated that the down-regulation of the *FAD2-1A* and *FAD2-2B* genes triggered by the transcription factor *GmWRI14* is the underlying mechanism reducing the LA level of seeds. These results could help geneticists and breeders to better understand the deposition of LA in soybean seeds. Meanwhile, this study also contributes key clues, providing further illumination of the regulatory mechanism of fatty acid accumulation.

## Figures and Tables

**Figure 1 ijms-23-00454-f001:**
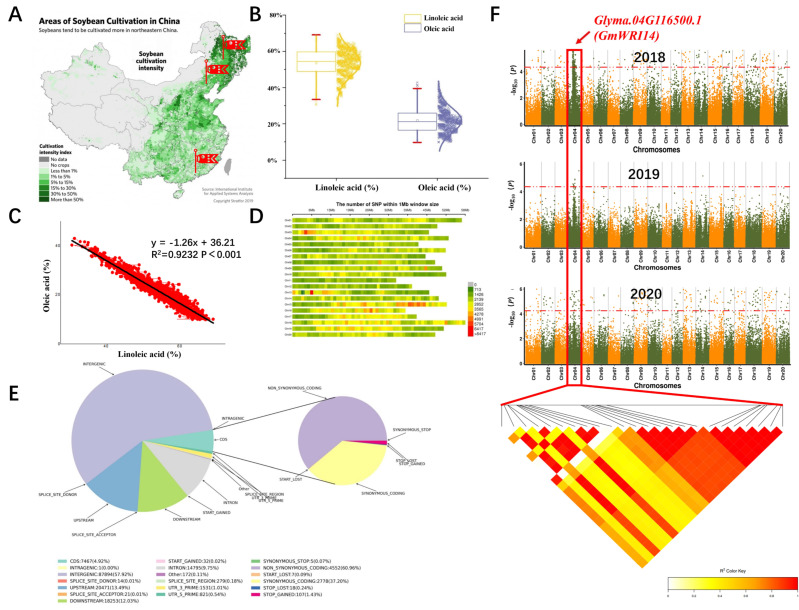
Phenotyping and genotyping of 510 soybean germplasms. (**A**) A total of 510 soybean germplasms from China (Heilongjiang province, Shandong province, and Guangdong province) were collected as the natural population. (**B**) The distribution of the LA content of different soybean lines resembled the bell-shaped curve of a normal distribution. (**C**) The association between the percentages of oleic acid and LA, y = −1.26x + 36.21, is a linear equation, where x represents the LA content and y represents the oleic acid content. (**D**) Distribution map of SNPs on different chromosomes. The abscissa is the length of the chromosomes. Each band represents one chromosome. (**E**) Pie chart of SNP annotations. Left panel: SNP percentages generally associated with genes. (**F**) Genome-wide Manhattan plots of associations for LA content for a 2018–2020 analysis. In the left panel, the *x*-axis indicates the SNPs along each chromosome; the *Y*-axis indicates the −log_10_ (*p*-value) for the association.

**Figure 2 ijms-23-00454-f002:**
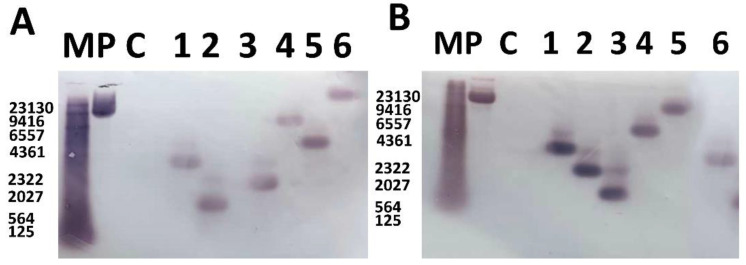
Verification of *GmWRI14* expression in T0/T1 transgenic soybeans using Southern blot analysis. (**A**) Southern blot analysis of the copy number of the *GmWRI14* expression cassette in T0 plants. (**B**) Copy number of the *GmWRI14* expression cassette in T1 plants. P: positive control. C: control JN38. M: marker. The full-length original blot has been included in the additional files Appendix A.

**Figure 3 ijms-23-00454-f003:**
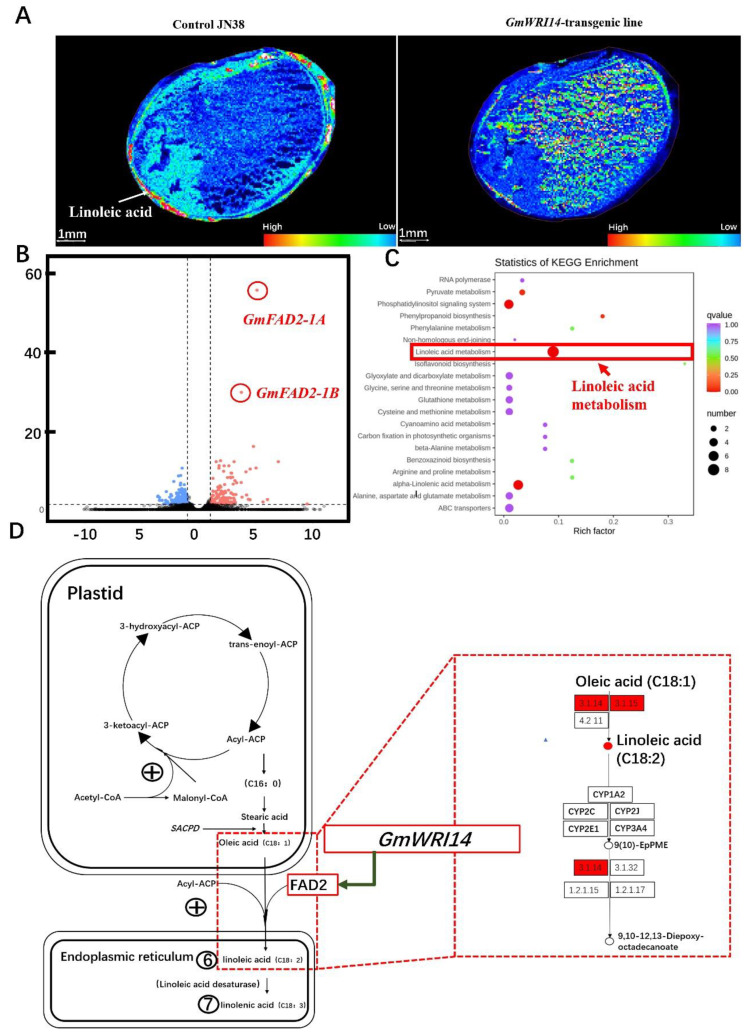
DEGs related to LA in *GmWRI14* transgenic soybean plants. (**A**) Analysis of the LA distribution in soybean seed using MALDI-TOF-MS. The red region represents LA. Scale bar, 100 mm. (**B**) DEGs were screened by transcriptomics. The red dot represents the up-regulated genes, while the blue dot represents the down-regulated differential genes. (**C**) Functional characterization of *GmWRI14* transgenic soybean for a KEGG enrichment analysis. (**D**) DEGs of a KEGG pathway map. For the *GmWRI14* transgenic soybean, the enzyme labeled with a red box was related to the down-regulated gene. The number in the box represents the fatty acid dehydrogenase (enzyme number), which explains the origin of phenotypic differences through the pathway.

**Figure 4 ijms-23-00454-f004:**
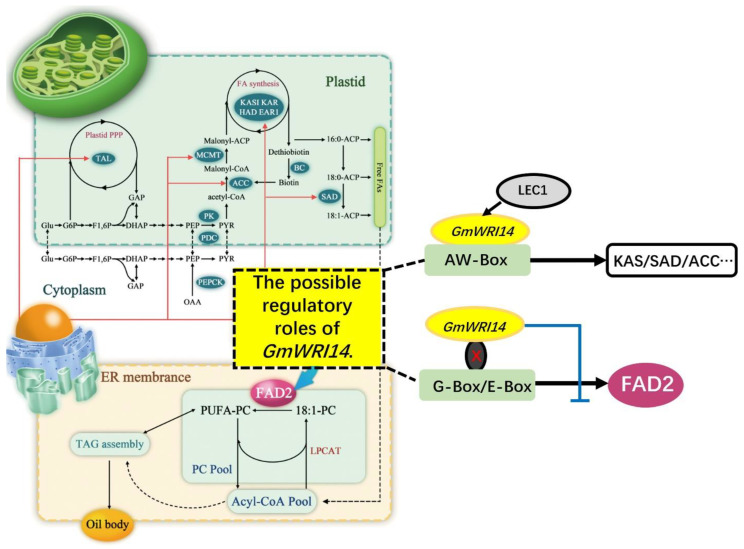
The model depicting transcriptional control by *GmWRI14*. Red arrow lines represent the possible regulatory roles of *GmWRI14*. Blue bar: *GmWRI14* suppresses the expression of *FAD2* gene by cooperating with the trans-acting factor (depicted as “X”). Abbreviations: PEPCK, phosphoenolpyruvate carboxykinase; SAD, stearoyl-ACP desaturase; BS, biotin synthase; KAS, ketoacyl-ACP synthase; EAR1, enoyl-ACP reductase 1; KAR, ketoacyl-ACP reductase; MCMT, malonyl-CoA; BC, biotin carboxylase; TAL, transaldolase; FAD2, fatty acid desaturase 2. Black arrow in G-Box/E-Box: G-Box/E-Box elements regulate *FAD2* gene transcription; blue bar in G-Box/E-Box: *GmWRI14* suppresses the expression of *FAD2* genes by interfering with E-Box/G-Box-mediated transcription.

**Table 1 ijms-23-00454-t001:** Analysis of *GmFAD2-1A* and *GmFAD2-2B* expression in different tissues of transgenic soybeans.

Gene	Correlation Coefficient with WRI1	Soybean Name	Relative Expression in Leaves		Relative Expression in Stem		Relative Expression in Roots		Relative Expression in Seed	
			Mean	Sig.	Mean	Sig.	Mean	Sig.	Mean	Sig.
** *GmFAD2-1A* **	−0.970–−0.982	JN38	10.15 ± 0.34	c	36.11 ± 0.42	b	32.21 ± 0.12	a	35.11 ± 0.25	a
		JN38-*GmWRI14*-1	5.31 ± 0.24	d	14.23 ± 0.21	c	13.23 ± 1.2	c	19.23 ± 0.54	c
		JN38-*GmWRI14*-2	2.52 ± 0.32	d	10.21 ± 0.51	c	23.23 ± 0.12	b	17.32 ± 0.42	c
		JN38-*GmWRI14*-3	3.12 ± 0.01	d	19.23 ± 0.15	c	29.11 ± 0.51	b	22.23 ± 0.56	b
		JN38-*GmWRI14*-4	4.25 ± 0.12	d	15.22 ± 0.16	c	16.23 ± 0.23	c	26.11 ± 0.16	b
		JN38-*GmWRI14*-5	3.22 ± 0.02	d	18.25 ± 0.31	c	20.54 ± 0.26	b	23.22 ± 0.61	b
		JN38-*GmWRI14*-6	1.33 ± 0.01	d	18.22 ± 0.25	c	21.51 ± 0.16	b	19.23 ± 0.12	c
** *GmFAD2-2B* **	−0.880–−0.814	JN38-*GmWRI14*-1	6.23 ± 0.21	d	3.14 ± 0.17	d	11.95 ± 0.54	c	22.75 ± 0.26	b
		JN38-*GmWRI14*-2	5.46 ± 0.11	d	4.22 ± 0.22	d	12.51 ± 0.34	c	21.21 ± 0.13	b
		JN38-*GmWRI14*-3	6.45 ± 0.15	d	4.42 ± 0.2	d	11.11 ± 0.54	c	26.12 ± 0.26	b
		JN38-*GmWRI14*-4	7.23 ± 0.17	d	5.12 ± 0.15	d	13.21 ± 0.32	c	25.15 ± 0.65	b
		JN38-*GmWRI14*-5	7.51 ± 0.22	d	4.78 ± 0.25	d	9.51 ± 0.26	d	27.35 ± 0.43	b
		JN38-*GmWRI14*-6	9.11 ± 0.12	d	3.22 ± 0.19	d	12.43 ± 0.62	c	26.21 ± 0.22	b

Note: The different lower letters indicate significant differences at *p* < 0.05, as determined by Duncan’s multiple-range test.

## Data Availability

The data that support the findings of this study are available from (Beijing Biomarker Biotechnology Co., Ltd., Beijing, China), but restrictions apply regarding the availability of these data, which were used under license for the current study and so are not publicly available. Data are, however, available from the authors upon reasonable request and with the permission of (Beijing Biomarker Biotechnology Co., Ltd., Beijing, China). Other datasets supporting the conclusions of this article are included within the article and its additional files.

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
