# Peer review of "Genome-Wide Association Study Identifies Candidate Genes Related to the Linoleic Acid Content in Soybean Seeds"

_ijms, 2021, doi:10.3390/ijms23010454_

Round 1
Reviewer 1 Report
The study exploited a soybean natural population consisted of 510 accessions to investigate the genetic architecture of linoleic acid in soybean seed. The study also reported that the downregulation of FAD2 gene regulated by a bunch of transcription factor WRI1 members is controlling the reduced linoleic acid level in soybean seeds. Data are presented adequately and there is a flow in the delivery of information. I strongly recommend authors to make extensive English editing for the whole manuscript.
Specific comments
Abstract
- Please double check for extra spaces between words.
- Line: 11-12: usually composition of soybean fatty acid profile is ordered as follow: palmitic, stearic, oleic, linoleic, and linolenic.
- Line: 16: the sentence needs to be edited to either be “After a total of 2,423,512 single-nucleotide polymorphisms (SNPs) were…….” or “After a total of 2,423,512 single-nucleotide polymorphism (SNP) markers were……”.
- Line: 18:genome-wide association analysis not Genome-wide association analysis (GWAS).
- Line: 19:We not we.
Introduction
- Line: 35-38: a very long sentence, it needs to be shortened and paraphrased carefully.
- Line: 53: “have been identified” not “have identified”.
- Line: 62-64: a very long sentence, it needs to be shortened and paraphrased carefully.
- Line: 69: introduction not Introduction
- Line: 69: “Clearly, WRI1 acts as a key” not “Clearly WRI1acts as a key” take care of extra spaces and comma needed.
- Line: 74: there is no “The” before “genome wide association analysis”, start the sentence by “Genome wide association analysis…….”
- Line: 75: add “single nucleotide polymorphisms” before the abbreviation “SNPs”
- In the last paragraph add some information about population you manipulated and regions where you collected from and which sequencing technology you used “ I think u=in your case its Specific-locus amplified fragment sequencing (SLAF-seq)”.
Materials and Methods
General comment: English editing is needed for the whole “Materials and Methods”
Section.
Line: 385: please add reference.
Line: 388: it is already well known for all researchers in molecular sciences that “DNA extraction is the first step in sequencing” so no need to this sentence, in addition you already mentioned the DNA extraction step in the first line.
Line 382-388: the whole paragraph needs comprehensive scientific rewriting.
Line 394: mistyping in “10-6”, change it to 10-6 .
Line 394: “Manhattan map was…… “ not were, otherwise, “Manhattan maps were…… .
Line 396: “genotypic data” not “genotype data…”
Line 419-420: Please link the two sentences together.
Line 446: Each not each.
Line 449: what do mean by “The positive and negative maps” ?
Results
Line 88-89 “was analyzed by SPSS 22.0 software” you already mentioned that in Materials and Methods not in Results.
Line 89 : standard deviation not Standard Deviation.
Line 93 : Why you said “ multiple linear regression”, did you assume the regression of multiple fatty acids as independent variables on only linoleic as dependent variable?. It is just simple regression between two factors X “linoleic acid” and Y “ oleic acid”. Please reconsider that carefully.
Line 95: Again like above comment.
Line 97: Its only one correlation coefficient between linoleic and oleic acids, not so many coefficients.
Line 111: soybean not Soybean.
In (Figure 1B): Oleic acid and Linoleic acid not oleic acid and linoleic. First letters are capital of x axis labels.
In (Figure 1B): Please add the concentration as “%” to both oleic and linoleic acids on X axis.
For example, Oleic acid (%).
In (Figure 1C): Linoleic acid should be on X-axis not on Y-axis due to being a dependent variable, and vice versa with oleic acid. And change this in the text in lines 114-115.
In (Figure 1C): make the first letter in both fatty acids capital not small case, and also add “%” to both fatty acids as stated above.
In (Figure 1C): To make this figure`s result more scientifically and statically accurate, each of linear equation, r value, and p-value for such relationship should be indicated on the plot itself in the figure.
Line 116: chromosomes not chromosome.
Line 118: Are you really sure that this Manhattan plot is for oleic acid ????, please be careful.
Line 125: was not were
Line 129: “the Manhattan for linoleic acid” add maps or plots to Manhattan.
Line 131: Again SO STRANGE MISTAKE, are you sure “genes related to soybean oleic acid content”???.
Line 135: Table A1 NOT Table.A1
Line 145: “the GmWRI14 expression was measured in………..
Line 150, 152: Please unify the way of writing the table number even if it was supplementary data.
Line 151: “between GmWRI14 expression level and linoleic acid content….”
Line 151: To make this sentence more scientifically and statically accurate, please rewrite it again carefully.
Line 158: please put the Agrobacterium tumefaciens in italic case.
Line 159-161: please clarify this sentence.
Line 165: clarify the structure either use “additional files in Figure A2 or additional files (Figure A2).
Line 172, 182: add comma after Meanwhile
Line 180: a significant not an significant
Line 200: not suitable expression “no-transgenic soybean.
Line 202: 5.1×107
Line 204: were screened by……
Line 210: delete are…
Line 216: Meanwhile,…
Line 219: As general conclusion,…
Line 219-221: To make this sentence more scientifically and statically accurate, please rewrite it again carefully.
Table 1: where are the uppercase letters in Table 1, if there is no, please delete its note below the table.
Line 257: please clarify this sentence “KEGG analysis linoleic acid metabolism”.
Discussion
Line 268-271: please paraphrase or “shorten” this sentence to make sense , and add suitable reference.
Line 282-285: please paraphrase to make sense.
Line 291-292: not clear sentence.
Line 297-301 : So long sentence , please shorten.
Line 302 : You always forget that your recent work is about linoleic not oleic acid.
Line 303: We introducing?????

Author Response
Dear reviewer
Thank your very much for your comments and suggestions, I am very sorry about the mistakes, and we followed your comments, the English language and style of revision manuscript had been re-edited by editing service, meanwhile, we added more information to the methods and results sections. Next I will answer the comments point-by-point.
Point 1: Line: 11-12: usually composition of soybean fatty acid profile is ordered as follow: palmitic, stearic, oleic, linoleic, and linolenic. 

Response 1: I followed your suggestion, the mistake have been corrected, the sentence is ‘five fatty acids (palmitic, stearic, oleic, linoleic, and linolenic)’
Point 2: Line: 16: the sentence needs to be edited to either be “After a total of 2,423,512 single-nucleotide polymorphisms (SNPs) were…….” or “After a total of 2,423,512 single-nucleotide polymorphism (SNP) markers were……”.
Response 2: the abstract has been re-edited. The corrected sentence is ‘After a total of 2,423,512 single-nucleotide polymorphism (SNP) markers’. Meanwhile the sentence have been moved to lins129
Point 3: Line: 18: genome-wide association analysis not Genome-wide association analysis (GWAS).

Response 3: I followed your suggestion, the mistake have been corrected, please check them. Line14
Point 4: Line: 19:We not we.

Response 4: The mistake have been corrected.
Point 5: Line: 35-38: a very long sentence, it needs to be shortened and paraphrased carefully.

Response 5: I followed your suggestion, the sentence has been shortened, the sentence is‘ linolenic acid can lower the cholesterol content in the blood [5,6], but it is not resistant to high temperatures, is not easy to preserve, and lowers the nutritional value of soybean oil [7–9]. Line 38-40.
Point 6: Line: 53: “have been identified” not “have identified”.
Response 6: I am sorry about the mistake, the mistake have been corrected, please check them. Line 54
Point 7: Line: 62-64: a very long sentence, it needs to be shortened and paraphrased carefully.

Response 7: the sentence have been corrected, please check them.‘WRINKLED1 (WRI1) encodes an APETALA2/ethylene responsive element binding 62 protein (AP2/EREBP). WRI1 is the key transcription factor found to directly regulate fatty 63 acid synthesis [22,23].’line 62-64
Point 8: Line: 69: introduction not Introduction

Response 8:The mistake have been corrected, please check them. Line: 67
Point 9: Line: 69: “Clearly, WRI1 acts as a key” not “Clearly WRI1acts as a key” take care of extra spaces and comma needed.

Response 9: Thank you. the mistake have been corrected, the sentence is ‘Clearly, WRI1 acts as a key’line: 70
Point 10: Line: 74: there is no “The” before “genome wide association analysis”, start the sentence by “Genome wide association analysis…….”
Response 10: ‘the’ have been removed, please check it. Line: 72
Point 11: Line: 75: add “single nucleotide polymorphisms” before the abbreviation “SNPs”

In the last paragraph add some information about population you manipulated and regions where you collected from and which sequencing technology you used “ I think u=in your case its Specific-locus amplified fragment sequencing (SLAF-seq)”
Response 11: I followed your suggestion, we add “single nucleotide polymorphisms” before the abbreviation “SNPs” line 73. Meanwhile I added some information about population and the sequencing technology. Line: 84-89.
Point 12: General comment: English editing is needed for the whole “Materials and Methods”Section. 

Response 12: I am very sorry about the mistake, the revision manuscript has been checked by using MDPI's Author Services, please check them.
Point 13: Line: 385: please add reference.
Response 13: We have added reference. Please check them. Line:407
Point 14: Line: 388: it is already well known for all researchers in molecular sciences that “DNA extraction is the first step in sequencing” so no need to this sentence, in addition you already mentioned the DNA extraction step in the first line.
Response 14: the mistake have been corrected, “DNA extraction is the first step in sequencing” have been removed from the sentence, please check them. Line:408
Point 15: Line 382-388: the whole paragraph needs comprehensive scientific rewriting.
Response 15: I am sorry about the mistake, the whole paragraph have been corrected, please check them. Line:405-414
Point 16: Line 394: mistyping in “10-6”, change it to 10-6 .
Response 16: the mistake have been corrected, please check them. Line:420
Point 17: Line 394: “Manhattan map was…… “ not were, otherwise, “Manhattan maps were…… .

Response 17: The mistake have been corrected, the corrected sentence is “Manhattan maps were……please check them. Line:421
Point 18: Line 396: “genotypic data” not “genotype data…”
Response 18: The mistake have been corrected, please check them. Line: 422
Point 19: Line 419-420: Please link the two sentences together.

Response 19: I followed your suggestion, we have linked the two sentences together. Line:449-450
Point 20: Line 446: Each not each. Line 449: what do mean by “The positive and negative maps” ?
Response 20: the mistake have been corrected, the sentence “The positive and negative maps” have been deleted, please check them. Line 465 and line 468.
Point 21: Line 88-89 “was analyzed by SPSS 22.0 software” you already mentioned that in Materials and Methods not in Results.
Response 21: the sentence had been removed from the section.
Point 22: Line 89 : standard deviation not Standard Deviation. Line 93 : Why you said “ multiple linear regression”, did you assume the regression of multiple fatty acids as independent variables on only linoleic as dependent variable?. It is just simple regression between two factors X “linoleic acid” and Y “ oleic acid”. Please reconsider that carefully. Line 95: Again like above comment. Line 97: Its only one correlation coefficient between linoleic and oleic acids, not so many coefficients. Line 111: soybean not Soybean.
Response 22: I am sorry about the mistake (line 94). The sentence should be ‘We used the linear regression model to analyze the relationship between LA and oleic acid.’ please check them. Line98-99 , and the corrected sentence is ‘The correlation coefficient between oleic acid and LA was -0.750.’ line:101. The mistake have been corrected.
Point 23: In (Figure 1B): Oleic acid and Linoleic acid not oleic acid and linoleic. First letters are capital of x axis labels. In (Figure 1B): Please add the concentration as “%” to both oleic and linoleic acids on X axis. For example, Oleic acid (%).
Response 23: I followed your comments, the mistake have been corrected, First letters are capital of x axis labels. (Figure 1B). I add the concentration as “%” to both oleic and linoleic acids. please check them.
Point 24: In (Figure 1C): Linoleic acid should be on X-axis not on Y-axis due to being a dependent variable, and vice versa with oleic acid. And change this in the text in lines 114-115.
In (Figure 1C): make the first letter in both fatty acids capital not small case, and also add “%” to both fatty acids as stated above.
In (Figure 1C): To make this figure`s result more scientifically and statically accurate, each of linear equation, r value, and p-value for such relationship should be indicated on the plot itself in the figure.
Response 24: I followed your suggestion, Linoleic acid is on X-axis , I add the concentration as “%” to both oleic and linoleic acids . meanwhile, each of linear equation, r value, and p-value for such relationship have been indicated on figure 1C, please check them.
Point 25: Line 116: chromosomes not chromosome. Line 118: Are you really sure that this Manhattan plot is for oleic acid ????, please be careful. Line 125: was not were.Line 129: “the Manhattan for linoleic acid” add maps or plots to Manhattan. Line 131: Again SO STRANGE MISTAKE, are you sure “genes related to soybean oleic acid content”???.
Response 25: I am so sorry about all the mistakes, the mistakes had been corrected, please check them. Line 122, 124, 146,147.
Point 26: Line 135: Table A1 NOT Table.A1
Response 26: The mistakes had been corrected, please check them. Line 150.
Point 27:Line 145: “the GmWRI14 expression was measured in………..
Response 27: The mistakes had been corrected, Line 160.
Point 28:Line 150, 152: Please unify the way of writing the table number even if it was supplementary data.
Response 28: I am so sorry about all the mistakes, the mistakes had been corrected, please check them. Line 165, 167.
Point 29: Line 151: “between GmWRI14 expression level and linoleic acid content….”Line 151: To make this sentence more scientifically and statically accurate, please rewrite it again carefully.
Response 29: The mistakes had been corrected, lines 166. The corrected sentence is ‘As general conclusion, we can say that the correlation coefficient between GmWRI14 expression level and LA content is -0.901~-0.912 ’ please check them. Line 165-166.
Point 30: Line 158: please put the Agrobacterium tumefaciens in italic case. Line 159-161: please clarify this sentence.
Response 30 :I am so sorry about all the mistake, I mistakes had been corrected, please check them. Lines 173-176.
Point 31: Line 165: clarify the structure either use “additional files in Figure A2 or additional files (Figure A2). Line 172, 182: add comma after Meanwhile
Response 31: The mistakes had been corrected, please check them. line 177.185,194.
Point 32: Line 180: a significant not an significant. Line 200: not suitable expression “no-transgenic soybean. Line 202: 5.1×107. Line 204: were screened by……Line 210: delete are…Line 216: Meanwhile,…Line 219: As general conclusion,…
Response 32: The mistakes had been corrected, please check them. Line 193, 218, 218,220, 226,233,235.
Point 33: Line 219-221: To make this sentence more scientifically and statically accurate, please rewrite it again carefully.
Response 33: The sentence had been re-writed, line 235-239.
Point 34: Table 1: where are the uppercase letters in Table 1, if there is no, please delete its note below the table.
Response 34: We deleted the wrong sentence from the legends of Table 1.
Point 35: Line 257: please clarify this sentence “KEGG analysis linoleic acid metabolism”.
Response 35: The corrected sentence is ‘DEGs were screened by transcriptomics. The red dot represents the up-regulated genes, while the blue dot represents the down-regulated differential genes’.line 265.
Point 36: Line 268-271: please paraphrase or “shorten” this sentence to make sense , and add suitable reference.
Response 36: the corrected sentence is ‘Low LA soybeans have a competitive agronomic yield potential, with many candidate genes associated with LA and discovered by GWAS already having been reported [13,14].’ Line 278-279.
Point 37: Line 282-285: please paraphrase to make sense.
Response 37: the sentence is ‘In another study, the close correlation between WRI1 and Stearoy-Acyl-Carrier-Protein Desaturase (SAD) expression suggested the regulatory role of WRI1 in LA accumulation’ line 284-286.
Point 38: Line 291-292: not clear sentence.
Response 38: the sentence had been removed from the paragraph.
Point 39: Line 297-301 : So long sentence , please shorten.
Response 39:the sentence had been shorten, the corrected sentence is ‘In previous research, candidate SNPs and genes significantly associated with LA were screened by GWAS. The functions of the candidate genes involved in LA metabolism and regulation, such as transcription activator-like effector nucleases (TALENs), acetyl CoA carboxylase biotin carboxylase subunit, glycosyltransferase group 1, and FAD2,’line 300-303
Point 40: Line 302 : You always forget that your recent work is about linoleic not oleic acid.
Response 40 :I am so sorry about all the mistake, I had checked the whole MS. Thank you for your comments.
Point 41: Line 303: We introducing?????
Response 41: The corrected sentence should be ‘We introduced GmWRI14 into soybean’ line:308.
Thank your very much for your comments again, Have a nice day
Reviewer 2 Report
The present research article provides genetic and genomic information about linoleic acid content based on GWAS and further candidate gene identification and complementation study of identified candidate gene GmWRI14. The topic is relevant to soybean fatty acid composition importance and also holds intriguing findings of differences in the content of linoleic acid.
However, this article is written very poorly, and the major concern is methods and results are incompletely described. Before it considers for publication in IJMS its must go through extensive revision.
Hence I would like to endorse for Major revision. Apart from this, there are more comments for the improvement of MS, and hope authors wish to address them.
Abstract
Abstract is the first section of the paper that most people read, and it should be used, sum up the entire study. Some components are missing in this case: What is the work's central premise? What are the most important outcomes? What contribution do the findings make to basic understanding? I would suggest rewriting the abstract.
The author mentioned that “510 soybean germplasms from China were collected as natural populations” What is this population? Is it a cultivated soybean or wild soybean, cultivar or varieties or inbred lines please explain.
Authors said “After a total of 2,423,512 nucleotide polymorphism (SNP) were obtained, a new candidate gene Glyma.04G116500.1 (GmWRI14) related to linoleic acid was discovered by 3- year long Genome-wide association analysis (GWAS)” This sentence is very vague and inappropriate.
Avoid words like “by 3 year long” instate use as “three consecutive years”
Introduction:
L 30-32, rewrite a sentence and the order of fatty acid should start with carbon (C16:0)-(C18:3).
Also can cite the recent article mentioned below, which showed different methods at the breeding genomics level for the fatty acid/ratio improvement.
“Krishnanand P. Kulkarni, Rupesh Tayade, Hyun Jo, Jong Tae Song and Jeong-Dong Lee (January 19th 2021). Breeding Strategy for Improvement of Omega-3 Fatty Acid through Conventional Breeding, Genetic Mapping, and Genomics in Soybean, Plant Breeding - Current and Future Views, Ibrokhim Y. Abdurakhmonov, IntechOpen, DOI: 10.5772/intechopen.95069. Available from: https://www.intechopen.com/chapters/74727”
Author can think about writing linoleic acid as (LA) and linolenic acid which is known as alpha (α) linolenic acid (ALA) as these words are repeated several times in the MS.
L 53, “linoleic acid have identified” it should be “linoleic have been identified”
L 54, “FAD2” should be defined with the complete name at first instance in the introduction and then use as FAD2.
L 59-60, revise sentence with “ Research showed ….”
L 63, define “APETALA2, OLE” with the complete name.
L 70, At some places transcription factor WRINKLED1 (AtWRI1), is written in italic please check throughout the MS and maintain the consistency, Similarly, gene names are not italic at several instances throughout the MS, please correct it.
There is nothing written about the GWAS (fastlmmc) model used for this study. Like why you chose this any advantage or importance, previous studies which used, etc..? It would worth including aspects in the Introduction part.
Results:
L86, Please modify the subtitle of “Phenotypic identification of linoleic acid content in soybean seeds” it should be “Phenotypic variation of linoleic acid content in soybean seeds”
GWAS results are not explained properly, figure 1F shows that there are several other chromosome regions also associated with the trait why the author does not select other regions for identification of candidate genes, and what criteria used for threshold level are not mentioned.
Please check the strain name mentioned “Agrobacterium tumefaciens” throughout the MS it should be written in italic.
L225, Figure is written in italic at some instance please check and correct and maintain the uniformity.
Figure 3. “Analysis different expression genes (DEGs)” should be written as “Analysis of differential expression genes”
There is no descriptive statistic for three years GWAS, please explain.
Materials and Methods:
It would be necessary to give information about the germplasm that was used. Why was the population structure analysis, not carried out as the authors used huge germplasm for the study?
L369-374, “The field was divided into three blocks and those were subdivided into eight sections. After that we did natural drying, then seeds were threshed for linoleic acid determination” modify it's not written properly, directly after plotting author mentioned natural drying, then seeds were threshed for linoleic acid determination. Please explain at what stage (R6, R7,…) seeds were harvested then harvested seeds were used for phenotyping for LA acid content, and describe similarly further. Otherwise, this method looks incomplete.
L375, 4.2. Determination of fatty acids in soybean Seeds: The method needs to explain in detail the process. How one should know what is NIRSTM DS 2500 (FOSS, Hillerod, Denmark) what steps are followed. Please explain.
L 384, “Specific-Locus Amplified Fragment Sequencing” no need to write the complete name as it appears before section define where it appears at first and then use (SLAF-Seq).
L389, GWAS section:
How many markers were obtained after the genotyping, all the markers obtained from genotyping were used for GWAS, what was the threshold used, what filtering criteria were used for analysis please explain.
L424, “4.7. Analysis of linoleic acid distribution in seed” please explain is its distribution or quantification or detection of content? What is different in “4.2 Determination of fatty acids in soybean Seeds” and 4.7 section. I guess it should be combined and written properly.
Two different versions of SPSS mentioned in the method why please explain?
Author did not explain anything using what method they identified the candidate gene for LA. The Similarly, transformation method was not described properly please explain why?
Figure 4. G-Box/E-Box, showing (suppression) red arrow and enhance (black) arrow it's confusing what exactly happing?
Discussion:
All the abbreviations used not defined previously and used as abbreviations please correct them. GWAS results not discussed properly with previously studied as there are several reports identified the QTL/SNPs and GWAS QTL in soyabase (database) for the LA how do your results correlate with previous studies please explain. The region identified for the LA is colocalizing with previously studied or not?
Conclusions:
Need to conclude the study properly what authors find and how? Re-write the conclusion.
Author Response
Dear reviewer
Thank your very much for your comments and suggestions, I am very sorry about the mistakes, and we followed your comments, the English language and style of revision manuscript had been re-edited by editing service, meanwhile, we added more information to the methods and results sections. Next I will answer the comments point-by-point.
Point 1: Abstract is the first section of the paper that most people read, and it should be used, sum up the entire study. Some components are missing in this case: What is the work's central premise? What are the most important outcomes? What contribution do the findings make to basic understanding? I would suggest rewriting the abstract. The author mentioned that “510 soybean germplasms from China were collected as natural populations” What is this population? Is it a cultivated soybean or wild soybean, cultivar or varieties or inbred lines please explain. Authors said “After a total of 2,423,512 nucleotide polymorphism (SNP) were obtained, a new candidate gene Glyma.04G116500.1 (GmWRI14) related to linoleic acid was discovered by 3- year long Genome-wide association analysis (GWAS)” This sentence is very vague and inappropriate. Avoid words like “by 3 year long” instate use as “three consecutive years”
Response 1: Thank you so much for your advice and help, I am very sorry about my mistakes, I followed your suggestions, the abstract section have been re-edited, I added more information about the population, it is cultivar, meanwhile, the mistakes had been corrected, please check them, abstract section: line 11-28.
Point 2: L 30-32, rewrite a sentence and the order of fatty acid should start with carbon (C16:0)-(C18:3). Author can think about writing linoleic acid as (LA) and linolenic acid which is known as alpha (α) linolenic acid (ALA) as these words are repeated several times in the MS.
Response 2: I am sorry about my mistakes, the sentence have been re-write, the corrected sentence is ‘Soybean oil is a complex mixture of five fatty acids: palmitic acid (C16:0), stearic acid (C18:0), oleic acid (C18:1), linoleic acid (LA) (C18:2), and linolenic acid (C18:3)’please check them. Line33-35. Meanwhile, follow your suggestions, we use LA to represent linoleic acid.
Point 3: L 53, “linoleic acid have identified” it should be “linoleic have been identified”
Response 3: The mistake has been corrected, please check them. Line:54.
Point 4: L 54, “FAD2” should be defined with the complete name at first instance in the introduction and then use as FAD2.

Response 4: The mistake have been corrected. Line:54.
Point 5: L 59-60, revise sentence with “ Research showed ….”
Response 5: I followed your suggestion, the sentence has been corrected, the corrected sentence is ‘Research showed that the loss of function mutations seen in FAD3 result’ line 59.
Point 6: L 63, define “APETALA2, OLE” with the complete name.
Response 6: The mistake have been corrected, the sentence is ‘ WRINKLED1 (WRI1) encodes an APETALA2/ethylene responsive element binding protein (AP2/EREBP).’please check them. Line 62. Meanwhile, because the OlE is the name of promoter, so the corrected sentence is ‘ZmWRI1 expressed under the embryo-preferred OLE promoter increased’ Please check them. line 66.
Point 7: L 70, At some places transcription factor WRINKLED1 (AtWRI1), is written in italic please check throughout the MS and maintain the consistency, Similarly, gene names are not italic at several instances throughout the MS, please correct it.
Response 7: I am very sorry about my mistakes. The mistakes had been corrected, I am so sorry.
Point 8: There is nothing written about the GWAS (fastlmmc) model used for this study. Like why you chose this any advantage or importance, previous studies which used, etc..? It would worth including aspects in the Introduction part.
Response 8: We added more information about GWAS (fastlmmc) model used for this study the. The sentence is ‘As factored spectrally transformed linear mixed models (fastlmmc model) run an order of magnitude faster than other efficient algorithms, the fastlmmc model was used for GWAS in many studies.’ Line: 76-77.
Point 9: L86, Please modify the subtitle of “Phenotypic identification of linoleic acid content in soybean seeds” it should be “Phenotypic variation of linoleic acid content in soybean seeds”
Response 9: Thank you for your Comments, the subtitle is “Phenotypic variation of LA content in soybean seeds” line: 91.
Point 10: GWAS results are not explained properly, figure 1F shows that there are several other chromosome regions also associated with the trait why the author does not select other regions for identification of candidate genes, and what criteria used for threshold level are not mentioned.
Response 10: We followed your suggestion, we added more information in the GWAS results section. The sentence is ‘The threshold value was 130 set at –log(p) > 4.20 (red). A total of 612 SNPs of LA were detected on chromosomes 1, 4, 131 9, 13, 14, 15, 16, 19, and 20. Among these, more than 50% of the SNPs were located in the 132 intergenic regions (the stretch of DNA sequences located between genes). In total, 4.98% 133 of the SNP loci were located in protein coding regions. Additionally, 6.34% of the SNP 134 markers were located in introns (Figure 1E). We found that most of the SNPs on chromosome 4 were exactly located at the coding sequence coordinates (CDS):’ line:130-135.
Point 11: Please check the strain name mentioned “Agrobacterium tumefaciens” throughout the MS it should be written in italic.
Response 11: I am very sorry about my mistakes, I have check the mistakes throughout the MS.
Point 12: Figure 3. “Analysis different expression genes (DEGs)” should be written as “Analysis of differential expression genes” here is no descriptive statistic for three years GWAS, please explain.
Response 12: the sentence has been corrected. Line 263. and I added more information of descriptive statistic. Line:129-135.
Point 13: It would be necessary to give information about the germplasm that was used. Why was the population structure analysis, not carried out as the authors used huge germplasm for the study?
Response 13: Thank you for your comments , I added more information about the germplasm( line 84-89), meanwhile, I added the population structure analysis (line138-142), and the Figure A1 is the result of population structure analysis.
Point 14: L369-374, “The field was divided into three blocks and those were subdivided into eight sections. After that we did natural drying, then seeds were threshed for linoleic acid determination” modify it's not written properly, directly after plotting author mentioned natural drying, then seeds were threshed for linoleic acid determination. Please explain at what stage (R6, R7,…) seeds were harvested then harvested seeds were used for phenotyping for LA acid content, and describe similarly further. Otherwise, this method looks incomplete.
Response 14: The mistake have been corrected, “The field was divided into three blocks and those were subdivided into eight sections. During late R8 soybean growth stages, the seeds were harvested and then the harvested seeds were used for phenotyping for the LA content’ line380-383.
Point 15: L375, 4.2. Determination of fatty acids in soybean Seeds: The method needs to explain in detail the process. How one should know what is NIRSTM DS 2500 (FOSS, Hillerod, Denmark) what steps are followed. Please explain.
Response 15: The corrected sentence is ‘The content of LA and another four fatty acids (stearic acid, palmitic acid, oleic acid, and linolenic acid) in 510 soybean cultivars seeds was determined by NIRSTM DS 2500 (FOSS, Hillerod, Denmark). Fresh seeds of the R8 stages were ground into a fine powder in liquid nitrogen and we put the powder into the instrument. The near-infrared spectroscopy mathematical model of soybean fatty acid content was provided by the Beijing Biomarker Biotechnology Company, PR China’ line 385-389.
Point 16: L 384, “Specific-Locus Amplified Fragment Sequencing” no need to write the complete name as it appears before section define where it appears at first and then use (SLAF-Seq).
Response 16: the mistake has corrected throughout the MS.
Point 17: How many markers were obtained after the genotyping, all the markers obtained from genotyping were used for GWAS, what was the threshold used, what filtering criteria were used for analysis please explain.
Response 17: We added more information to describe the result of GWAS. The sentence is ‘Based on the 2,423,512 SNP markers obtained by SLAF-Seq technology, all the SNP markers obtained from genotyping were used for GWAS. The TASSEL software can cal culate the Q matrix of sample population structure according to the K matrix and finally obtain a correlation value for each SNP maker. The threshold value was set at –log(p) > 4.20.’ and ‘We evaluated the genome-wide LD in 510 accessions and found that the LD (R2) values decayed to half of the maximum value within 8.9 kb. Using 8.9 kb as the linkage disequilibrium attenuation distance, candidate genes related to soybean linoleic acid traits were screened within the LD distance.’ Line 416-428.
Point 18: L424, “4.7. Analysis of linoleic acid distribution in seed” please explain is its distribution or quantification or detection of content? What is different in “4.2 Determination of fatty acids in soybean Seeds” and 4.7 section. I guess it should be combined and written properly.
Response 18: We followed your suggestion, the 4.2 and 4.7 have been combined, and the LA distribution in the seed was detected 390 using MALDI-TOF-MS. Line:385-390.
Point 19: Two different versions of SPSS mentioned in the method why please explain?
Response 19: I am sorry about the mistakes, the versions of SPSS is the same, the mistake have been corrected, please check them.
Point 20: Author did not explain anything using what method they identified the candidate gene for LA. The Similarly, transformation method was not described properly please explain why?
Response 20: We add more information to the method, the sentence is ‘We evaluated the genome-wide LD in 510 accessions and found that the LD (R2) values decayed to half of the maximum value within 8.9 kb. Using 8.9 kb as the linkage disequilibrium attenuation distance, candidate genes related to soybean linoleic acid traits were screened within the LD distance.’ Line 425-428. meanwhile, we add more information to the transformation method. Line 445-455.
Point 21: Figure 4. G-Box/E-Box, showing (suppression) red arrow and enhance (black) arrow it's confusing what exactly happing?
Response 21: The more information was added to the legend of Figure 4, the sentence is ‘Black arrow in G-Box/E-Box: G-Box/E- Box elements regulate FAD2 gene transcription; red arrow in G-Box/E-Box: GmWRI14 suppresses the expression of FAD2 genes by interfering with E-Box/G-Box-mediated transcription’ line371-373.
Point 22: All the abbreviations used not defined previously and used as abbreviations please correct them. GWAS results not discussed properly with previously studied as there are several reports identified the QTL/SNPs and GWAS QTL in soyabase (database) for the LA how do your results correlate with previous studies please explain. The region identified for the LA is colocalizing with previously studied or not?
Response22: I followed your suggestion, we added more information to the discussion section, the sentence is ‘In previous research, candidate SNPs and genes significantly associated with LA were screened by GWAS. The functions of the candidate genes involved in LA metabolism and regulation, such as transcription activator-like effector nucleases (TALENs), acetyl- CoA carboxylase biotin carboxylase subunit, glycosyltransferase group 1, and FAD2, were identified [16-18]. In our work, GWAS was used to find potential SNPs and genes correlated with the LA content, with most SNPs on chromosome 4 being found to be exactly located at CDS region 11,65,00–12,28,90, the region identified for the LA is colocalizing with previously studied’ line 299-305.
Point 23: Need to conclude the study properly what authors find and how? Re-write the conclusion.
Response 23: I am very sorry about the mistake, the conclusion had been re-writed, the sentence is ‘In this study, a genome-wide association study (GWAS) was conducted on a panel of 510 germplasm resources. We combined the GWAS and RNA-seq methods to identify candidate genes for LA metabolism in soybean seeds. A new candidate gene, GmWRI14 was discovered by GWAS in three consecutive years. The RNA-seq results indicated that the down-regulation of the FAD2-1A and FAD2-2B genes triggered by the transcription factor GmWRI14 is the underlying mechanism reducing the LA level of seeds. These results could help geneticists and breeders to better understand the deposition of LA in soybean seed. Meanwhile, this study also contributes key clues, providing the further illumination of the regulatory mechanism of fatty acid accumulation.’ Line 471-479.
Thank your very much for your comments again, Have a nice day
Round 2
Reviewer 2 Report
I have still a few major concerns about this article
- The author said there is no function for gene04G116500 name in a soybean database that’s is correct but in the description, functional annotations represent as “ANKYRIN REPEAT PROTEIN” Please explain why and what basis you named it as “GmWRI14”?
- The transcription factor can not be italic on several occasions it's written in italic. Apart from that few errors are still there.
Introduction:
L 289, rewrite a sentence as “while GWAS was conducted for finding the genomic regions associated with LA. Besides, a total of…….candidate genes were identified related to LA”.
Results:
L612 please change the word “simultaneously” to “consistently”
Discussion:
L1109-1110 revise sentence its incomplete, it should be “led to increasing…..”
Author Response
Dear reviewer
Thank your very much for your comments and suggestion, I am very sorry about the mistakes, and we followed your comments, some sentences have been re-edited, meanwhile, we added more information and a new figure (Supplementary Files Figure A2). Next I will answer the comments point-by-point.
Point 1:
The author said there is no function for gene04G116500 name in a soybean database that’s is correct but in the description, functional annotations represent as “ANKYRIN REPEAT PROTEIN” Please explain why and what basis you named it as “GmWRI14”?
Response 1: Thank you so much for your advice and comments, I am very sorry about my mistake, I followed your suggestion, I added more information and (figure A2) about the sequence.
‘The amino acid sequence encoded by Glyma.04G116500.1 was compared with AtWRI1 (Gene ID: 824599), the homology of the amino acid sequences between Glyma.04G116500.1 and AtWRI1 is 62.34%. The alignment analysis showed that Glyma.04G116500.1 contained two (AP2/EREB) DNA-binding domains (Figure A2), which were located at positions 54 to 220 of the amino acid sequence. Therefore, it is speculated that Glyma.04G116500.1 may belong to AP2 / EREBP family. Meanwhile, according to Swissport annotation, the Glyma.04G116500.1 belongs to the plant WRI1 family. Because the Glyma.04G116500.1 is located on chromosome 4, so we named it as “GmWRI14”. Line:155-164.
Point 2: The transcription factor can not be italic on several occasions it's written in italic. Apart from that few errors are still there.
Response 2: I am sorry about my mistake, the mistake had been corrected, please check. They are Lines: 62, Lines :70, Lines :155, Lines :269, Lines :324, Lines :326.
Point 3: L 289, rewrite a sentence as “while GWAS was conducted for finding the genomic regions associated with LA. Besides, a total of…….candidate genes were identified related to LA”.
Response 3: the sentence had been re-write, the corrected sentence is ‘while GWAS was conducted for finding the genomic regions associated with LA. Besides, a total of 37 candidate genes were identified related to LA. A new candidate gene related to LA was discovered by GWAS in three years consistently.’ Line:88-90.
Point 4: L612 please change the word “simultaneously” to “consistently”
Response 4: The mistake have been corrected. Line:155.
Point 5: L1109-1110 revise sentence its incomplete, it should be “led to increasing…..”
Response 5: I followed your suggestion, the sentence has been corrected, the corrected sentence is ‘the introduction of the Arabidopsis transcription factor AtWRI1 into soybean plant led to increasing level of palmitate up to approximately 20%’ line 288